# The Hypoglossal Nerve Stimulation as a Novel Therapy for Treating Obstructive Sleep Apnea—A Literature Review

**DOI:** 10.3390/ijerph18041642

**Published:** 2021-02-09

**Authors:** Saif Mashaqi, Salma Imran Patel, Daniel Combs, Lauren Estep, Sonia Helmick, Joan Machamer, Sairam Parthasarathy

**Affiliations:** 1UAHS Center for Sleep & Circadian Sciences and Division of Pulmonary, Allergy, Critical Care and Sleep Medicine, Department of Medicine, University of Arizona College of Medicine, Tucson, AZ 85724, USA; salmapatel@deptofmed.arizona.edu (S.I.P.); lestep@deptofmed.arizona.edu (L.E.); Sonia.Helmick@bannerhealth.com (S.H.); jmachamer@deptofmed.arizona.edu (J.M.); sparthasarathy@deptofmed.arizona.edu (S.P.); 2UAHS Center for Sleep & Circadian Sciences and Division of Pulmonary and Sleep Medicine, Department of Pediatrics, University of Arizona, Tucson, AZ 85724, USA; dcombs@peds.arizona.edu

**Keywords:** obstructive sleep apnea, hypoglossal nerve stimulation, Inspire, genioglossus, DISE, BMI, complete concentric collapse, children, CPAP

## Abstract

Obstructive sleep apnea (OSA) is a common sleep disorder that affects all age groups and is associated with many co-morbid diseases (especially cardiovascular diseases). Continuous positive airway pressure (CPAP) is the gold standard for treating OSA. However, adherence to PAP therapy has been a major challenge with an estimated adherence between 20% and 80%. Mandibular advancement devices (MAD) are a good alternative option if used in the appropriate patient. MAD are most effective in mild and moderate OSA but not severe OSA. Surgical options are invasive, not appropriate for severe OSA, and associated with pain and long healing time. Hypoglossal nerve stimulation (HGNS), or upper airway stimulation (UAS), is a novel therapy in treating moderate and severe degrees of OSA in patients who cannot tolerate CPAP therapy. We reviewed the MEDLINE (PubMed) database. The search process yielded 303 articles; 31 met the inclusion and exclusion criteria and were included. We concluded that hypoglossal nerve stimulation is a very effective and novel alternative therapy for moderate and severe OSA in patients who cannot tolerate CPAP therapy. Adherence to HGNS is superior to CPAP. However, more developments are needed to ensure the highest safety profile.

## 1. Introduction

Obstructive sleep apnea (OSA) is a prevalent condition affecting approximately three to seven percent of males and two to five percent of females in the adult population [1]. Untreated OSA is associated with multiple comorbidities including hypertension [1], diabetes mellitus [2], coronary artery disease [3], stroke [4], congestive heart failure [5], atrial fibrillation [6], and death [7]. Untreated OSA is also associated with decreased quality of life indicators for patients and excessive daytime sleepiness [8]. Obstructive sleep apnea has a large economic impact when left untreated [9].

Obstructive sleep apnea is characterized by repetitive upper airway collapse during sleep [10]. Upper airway patency is maintained by contraction of the upper airway dilator muscles [10]. There are several upper airway dilator muscles [10]. The most important upper airway dilator muscle is the genioglossus muscle which has phasic activity during inspiration. The genioglossus muscle is innervated by the hypoglossal nerve (cranial nerve XII) [10]. Other dilator muscles such as the tensor palatini stiffen the upper airway by having tonic activity throughout inspiration and expiration [10]. The upper airway dilator muscles are generally effective in maintaining upper airway patency except when limited by anatomy (e.g., retrognathia [11]) or excessive loading from soft or fatty tissue [12]. Ensuring upper airway patency is the underlying mechanism of action of all OSA treatment methods [13]. 

There are many options for treating obstructive sleep apnea including weight loss, positional therapy (i.e., encouraging side sleeping), mandibular advancement devices, positive airway pressure therapy (PAP), and surgery [13]. As can be expected, weight loss is difficult to achieve and usually takes several years prior to being successful [14]. Positional therapy, effective only in a subset of patients, can be uncomfortable and often difficult for patients to adhere to [15]. Mandibular advancement devices may only be effective in mild and moderate cases of OSA and need the availability of a qualified dentist to fabricate the device [13,16]. Positive airway pressure therapy is very effective, however not well tolerated by many patients [13,16]. Finally, conventional surgical options including septoplasty, nasal polypectomy, adenoidectomy, tonsillectomy, uvulopalatopharyngoplasty, uvuloplasty, glossectomy, tongue base reduction, mandibular advancement, genioglossal advancement, hyoid myotomy suspension, maxillomandibular advancement, tracheostomy, and bariatric surgery can be quite invasive and have varying success rates depending on the surgery from 35 to 86% [17]. Upper airway stimulation (UAS) is a more recent surgical option for treating obstructive sleep apnea with a success rate of about 75% at 5 years [18]. In this review, we will focus on UAS using hypoglossal nerve stimulation (HGNS). We will review different devices used, mechanism of action, short- and long-term effectiveness, impact on cardiovascular diseases, the use of HGNS in children, adverse events, complications, and finally, the future of upper airway stimulation and challenges facing this field.

## 2. Materials and Methods

We reviewed the literature using the PubMed/MEDLINE database from 1980 to 2020. We used different phrases for the search process. Inclusion criteria: (A) Animal and human studies. (B) Studies published in English language. Exclusion criteria: (A) Non-clinical trial articles (e.g., review articles, editorials, and correspondence). (B) Articles that do not address the phrase used for search. This was completed by reviewing the abstracts. We used the following phrases: 1. “Hypoglossal nerve stimulation and obstructive sleep apnea in animals” and “Upper airway stimulation and obstructive sleep apnea in animals” which yielded 47 articles; 6 of them met the inclusion and exclusion criteria and subsequently, were included in the review. 2. “Hypoglossal nerve stimulation and obstructive sleep apnea in humans” and “Upper airway stimulation and obstructive sleep apnea in humans” which yielded 197 articles; 16 of them met the inclusion and exclusion criteria and subsequently, were included in the review. 3. “Hypoglossal nerve stimulation and cardiovascular diseases” and “Upper airway stimulation and cardiovascular diseases” which yielded 29 articles; 3 of them met the inclusion and exclusion criteria and subsequently were included in the review. 4. “Hypoglossal nerve stimulation children” and “hypoglossal nerve stimulation pediatric”. These terms yielded a total of 30 unique results; 6 results met the inclusion and exclusion criteria and were included in the review. As a result, a total of 31 articles were included in the review.

## 3. What Is a Hypoglossal Nerve Stimulator?

A hypoglossal nerve stimulator is a device that generates electrical impulses through a generator that is implanted in the upper right chest (under the skin). The impulse is transmitted via a tunneled lead that ends up with a cuff that surrounds the hypoglossal nerve. There are three HGNS devices that were tested in clinical trials. The Apnex Medical Inc. (St. Paul, MN, USA) device was promising in a phase II trial [19] but failed at phase III because it did not meet efficacy standards and the company does not exist now (Figure 1). The second device is the ImThera aura 6000 (San Diego, CA, USA). It places six electrodes around the trunk of the hypoglossal nerve [20,21]. It is still in phase III clinical trial (Figure 2). The third device, which is the only one approved by the FDA, is Inspire Medical Systems (Maple Grove, MN, USA) (Figure 3). Since it is the only one approved by the FDA, we will focus on the Inspire Medical System in our review.

HGNSs in general and Inspire in particular are composed of the following parts: A generator that contains batteries and generates the stimulus that is delivered to the hypoglossal nerve (Figure 4). The generator has two ports, one for the respiratory sensing lead and the other one for the stimulation lead. The respiratory sensing lead detects respiratory efforts (mechanical signal) and converts it into an electrical signal (Figure 5). The stimulation lead delivers the stimulation to the hypoglossal nerve via electrodes located in the inner surface of a self-sizing cuff (Figure 6). The remote is given to the patient after the activation process [22]. The following settings can be adjusted using the remote: turning therapy on and off, pausing therapy (when the patient wakes up during the night), and increasing/decreasing the stimulation strength (Figure 7) [23]. The programmer is a tablet that is used by the ENT surgeon intraoperatively or the sleep physician during the activation visit (Figure 8). The programmer allows the setting of different variables of the Inspire system [24]. 

## 4. Implantation Process (Surgical Technique), Pre- and Post-Implant Procedures, and Follow Up

The current FDA-approved Inspire device is placed during what is typically an outpatient procedure requiring general anesthesia that lasts between 90 and 180 min and requires three small skin incisions [25]. The first incision is placed typically in the right upper neck midline between the hyoid and mandible. After retracting the submandibular gland posterosuperior, the distal hypoglossal nerve is identified. The cuff is then placed around the nerve and irrigated with sterile saline. Cuff placement is tested using the generator or an external nerve stimulator, or recently, a nerve monitoring-guided selective hypoglossal nerve stimulator [26]. The second incision is made in the chest on the same side as the neck incision for the generator pocket where the pulse generator is implanted. The third incision is made on the same side for placement of a pleural respiratory sensor (Figure 9). All components are connected via leads that are subcutaneously tunneled. Electrical testing is performed at the time of surgery to ensure satisfactory tongue movement and adequate respiratory signal waveform before the incisions are closed. A postoperative chest X-ray is obtained for documentation of baseline device position and to rule out negative sequelae such as a pneumothorax. Such complications are rare, as the procedure does not directly involve the airway [27].

The role of sleep medicine providers is to identify OSA patients who have problems tolerating traditional CPAP therapy or CPAP failure. If they have an interest in HGNS, they are further screened with a complete health history, sleep comorbidities, and BMI. They undergo overnight diagnostic PSG (Polysomnography; this is discussed in detail in the next section).

If they meet the criteria, then the next step is referral to ENT where evaluation of the upper airway endoscopically is performed utilizing Drug-Induced Sleep Endoscopy (DISE), specifically looking for anteroposterior tongue base and palate collapse. Patients with complete concentric collapse at the level of the velum are poor candidates. Sleep providers also assist with titration of the device for optimal outcomes [28]. Following the implantation process, a follow up with ENT in four weeks is usually scheduled to assess for healing and any adverse events. During this time, the generator remains deactivated. Six weeks after implantation, the patient visits with a sleep provider to activate the generator (this process was discussed earlier in this review). Two months later, a full-night lab titration study is conducted to ensure the effectiveness of the Inspire settings (mainly functional threshold) in all sleep stages and in all sleep positions. Then, follow up visits at 3, 6, and 12 months can be scheduled to evaluate the long-term clinical outcome [29].

DISE is a preoperative requirement in patient selection for HGNS. The airway responds differently during wakefulness and sleep; during sleep, the upper airway has limited tone and muscle control. DISE mimics the airway during sleep, giving a more accurate picture of the degree of collapse and phenotype of collapsibility. Propofol, midazolam, and dexmedetomidine are the drugs of choice for DISE. The degree of sedation is important; a light sedation that most closely mimics sleep is ideal for accurate assessment of the upper airway in the context of selection of surgical candidates [30]. In patients with complete centric collapse (CCC), HGNS is associated with poor outcome [28].

## 5. The Clinical Indications, Contraindications, and Requirements for HGNS Consideration

Upper airway stimulation (UAS) is indicated in patients with moderate and severe OSA (AHI greater more or equal to 15 events per hour and less than or equal to 65 events per hour) who cannot tolerate or failed positive airway pressure (PAP) therapy. PAP failure is defined as persistent elevation in the apnea–hypopnea index (AHI ≥ 15 events per hour), while PAP intolerance is defined as the inability to use PAP therapy continuously (more than or equal to five nights per week for more than or equal to four hours every night) or the unwillingness to use PAP therapy again after quitting in the past. UAS is used in patients older than 22 years of age. However, it is still indicated in patients between 18- and 21-years-old, especially if adenotonsillectomy is contraindicated or they meet the previously noted criteria. Recently, the FDA approved using UAS in patients between 18 and 21 years [31].

UAS is contraindicated in patients with central sleep apnea (defined as a central apnea–hypopnea index of more than 25% of the total AHI), and in patients with sleep-related hypoxia or hypoventilation (such as patients with severe obstructive or restrictive pulmonary diseases). One of the factors that has been shown to be significantly correlated with the success of UAS is body mass index (BMI). BMI ≥ 32 kg/m^2^ is less likely to be associated with a successful outcome from UAS. The safety of UAS is still to be proven in pregnant patients and should not be attempted. As mentioned earlier in this review, one of the essential steps before considering UAS is the DISE exam. A complete concentric collapse (CCC) pattern of the palate during each apnea or hypopnea predicts a poor therapeutic success with UAS compared to palatal collapse in the anteroposterior axis [31].

HGNS (Inspire) should be considered carefully and cautiously in patients who require frequent Magnetic Resonance Imaging (MRI) scanning. Older Inspire models are not eligible for MRI use and the current Inspire is MRI conditional where they are compatible with MRI scanning of the upper/lower extremities, head, and neck only [32].

## 6. Mechanism of Action, Lessons from Animal and Human Studies Prior to the STAR Trial

In order to understand the mechanism of action of HGNS, we need to understand the mechanics of retropalatal, hypopharyngeal, and nasopharyngeal collapse and the role of tongue protrusion in alleviating this collapse. Tongue movement is controlled by two groups of muscles: (A) The extrinsic muscles that include genioglossus (GG) and geniohyoid (GH) muscles, which tend to protrude the tongue. The styloglossus (SG) and hyoglossus (HG) muscles usually tend to retract the tongue and oppose the action of GG and GH. (B) The intrinsic muscles, which are a group of horizontal fibers that tend to protrude the tongue anteriorly, and oblique and vertical fibers that tend to flatten the tongue (Figure 10) [33]. The pressure in the pharyngeal airway at which collapsibility ensues is called the pharyngeal critical closing pressure (Pcrit). The resistance upstream to the site of pharyngeal collapse is called the nasal resistance (Rn). In theory, increasing the maximum inspiratory airflow (VImax) during apnea decreases the pharyngeal collapsibility (Pcrit) and/or nasal resistance (Rn) (Appendix A).

Although, stimulating the tongue protrudor muscles (GG) and (GH) should improve the Pcrit and VImax, there is strong evidence from animal studies that this is true during expiration, while during inspiration, this causes more collapse. Moreover, stimulating both tongue protrudors and retractors causes pharyngeal dilatation during expiration and more airway stability. The hypoglossal nerve (CN XII) innervates both the tongue protrudors and retractors. Selective activation of the hypoglossal medial branch that innervates the tongue protrudors seems to be less effective in dilating the pharyngeal airway compared to co-activation of the hypoglossal nerve trunk or co-activation of the CN XII branch that innervates tongue retractor muscles (this is discussed thoroughly in the next section). In addition to tongue protrusion, it seems that the soft palate plays a major role also in maintaining retropalatal and hypopharyngeal patency and stiffens these structures through the contraction of the palatoglossus muscle (this is called palatoglossus coupling) [34,35,36]. This opening in the retropalatal space is variable from person to person, which can explain why sometimes, a good tongue protrusion results in a narrow opening [37]. Heiser et al. concluded in their paper that focused on palatoglossus coupling that bilateral stimulation of the hypoglossal nerve results in a better retropalatal opening in addition to tongue protrusion (i.e., better palatoglossus coupling) [36]. This facilitated the road for one of the newest modalities of UAS (this will be described in detail later in this review).

### 6.1. Animal Studies

The concept of hypoglossal nerve stimulation to decrease air way collapsibility was tested initially in animal models. Schwartz et al. [38] conducted the first test of HGNS on upper airway collapsibility in a cat model. They analyzed the pressure–volume relationships at different frequencies of HGNS from 0 to 100 Hz. They determined the maximum inspiratory airflow and the mechanical factors contributing to it, the upper airway critical pressure (Pcrit), and nasal resistance. They concluded that hypoglossal nerve stimulation increased the maximum inspiratory airflow by decreasing the Pcrit, and subsequently, decreased upper airway collapsibility, presumably secondary to tongue protrusion. Miki et al. [39] conducted a similar experiment on anesthetized dogs and concluded that hypoglossal nerve stimulation at a frequency higher than 50 Hz decreased upper airway resistance to a stable level, preventing upper airway collapse via stimulating the genioglossus muscle. As mentioned earlier in this review, the main trunk of the hypoglossal nerve innervates both the protrudor and retractor muscles of the tongue. However, the medial branch of the hypoglossal nerve innervates only the protrudor muscles. Bailey et al. [40] tested the difference in the independent stimulation of the protrudor muscles of the tongue and the co-activation of both (protrudor and retractor muscles) on pharyngeal airway size and upper airway stiffness in rats. They concluded that stimulation of the protrudor muscles dilated the pharyngeal airway at lower airway volume (<0.1 mL) but constricted the airway at higher volumes. However, co-activation had similar results at higher airway volumes but did not dilate the airway at lower volumes. Both independent stimulation of the protrudor muscles and co-activation increased airway stiffness, but this is a very complex process that mainly depends on airway volume. Similar results came from Yoo et al. [41] who tested the inspiratory flow and Pcrit in isolated canine upper airway. They used a special device that activates the whole trunk and selectively stimulates each branch of the hypoglossal nerve. Selective stimulation was tested for the branches that innervate GG and GH muscles (i.e., tongue protrudors) and branches that innervate SG and HG (i.e., tongue retractors). Nonselective activation included stimulation of the whole hypoglossal nerve trunk. Interestingly, they noticed that whole nerve activation or co-activation of protrudors (GG) and retractors (SH + HG) at the same time produced more stability of the upper airway in addition to more reduction in the Pcrit and more airway flow during inspiration compared to selective activation of the tongue protrudor (GH), which caused upper airway instability.

Oliven et al. [42] had similar results in anesthetized dogs when they directly stimulated the hypoglossal nerve and concluded a reduction in upper airway collapsibility and more dilatation and more stiffness via a decrease in upper airway resistance (from 9.0 to 0.3 cm H_2_O), which led to an increase in maximum inspiratory airflow (0.28 to 2.07 L/s) and an increase in Pcrit (from −2.7 to 2.1 cm H_2_O) (*p* < 0.002). Goding et al. [43] implanted an electrode cuff connected with a pulse generator and attached it to the hypoglossal nerve. The stimulation of the hypoglossal nerve increased the peak inspiratory air flow (from an average of 0.1 to 1.6 L/s) (*p* = 0.0001).

### 6.2. Human Studies

Schwartz et al. [44] tested direct intramuscular stimulation of lingual musculature and its impact on OSA in nine patients. All patients (eight males and one female) had severe OSA (mean NREM AHI 93 events per hour and mean REM AHI 80 events per hour). Selective stimulation of the tongue protrudors (GG) and retractors (HG and SG) was possible and that resulted in reduction in VI max (from 345 mL/s before HG and SG stimulation to 137 mL/s during stimulation) and increase in VI max (from 288 mL/s before GG stimulation to 501 mL/s during stimulation). The stimulation did not result in arousals (as demonstrated by EEG). The AHI was reduced from a mean of 65 events per hour without stimulation to 9 events per hour with stimulation.

Oliven et al. [45] had similar results, however, they had more detailed conclusions that compared the stimulation of different muscle fibers of GG to co-activation with HG. The study had two parts. The first part included seven patients diagnosed with OSA (AHI ≥ 5 events per hour) who underwent overnight PSG. Surface electrical stimulation of the GG muscle was conducted using only the anterior part of the sublingual electrode. This did not result in any tongue protrusion as reflected by no change in the nasal pressure (Pn): flow ratio and oropharyngeal cross-sectional area. Stimulation of the HG resulted in airway collapse. Co-activation of both (GG surface and HG) opened the airway as reflected by a shift of the Pn: flow ratio and increase in the oropharyngeal cross-sectional area. The second part of the study included seven patients with a diagnosis of OSA (AHI ≥ 5 events per hour) who underwent anesthesia using propofol infusion. Stimulation of the GG was conducted using intramuscular fine-wire electrodes which resulted in tongue protrusion as reflected by the Pn: flow ratio and increase in the cross-sectional area. Stimulation of the HG muscle caused obstruction and co-activation of combined muscles did not add additional dilatation compared to GG alone (mean cross-sectional area 0.171 cm^2^/cm H_2_O with GG stimulation compared to 0.171 cm^2^/cm H_2_O in combined co-activation).

The reason for these findings can be explained by the muscle fibers that constitute the GG muscle. There are vertical fibers that are spread in a fanlike orientation and cause flattening and depression of the tongue once stimulated but no protrusion (the fibers that are stimulated by surface electrical stimulation in patients underwent anesthesia). The other fibers are the longitudinal fibers that are located deeper in the tongue and cause protrusion of the tongue when stimulated.

Prior to the STAR trial, the landmark study of HGNS for OSA, several human trials were conducted to examine the impact of HGNS on polysomnographic variables and clinical symptoms of OSA. Eastwood et al. [46] conducted a prospective trial where HGNS (Apnex Medical Inc., St. Paul, MN, USA) was implanted in twenty-one patients with moderate–severe OSA who could not tolerate PAP therapy. In the whole group, the mean AHI was reduced to 19.5 events per hour from 43 events per hour at baseline compared to six months post implant. This was associated with significant improvement in event-associated oxygen desaturation, event-associated arousals, and spontaneous arousals. There was also improvement in the sleep architecture reflected by reduction in stage N and increase in REM sleep. Clinically, there was improvement in quality of life questionnaires (e.g., FOSQ (Functional Outcomes of Sleep Questionnaire), SAQLI (Sleep Apnea Quality of Life), and PSQI (Pittsburgh Sleep Quality Index)), and excessive daytime sleepiness scores (ESS). Moreover, HGNS was tolerated very well. Self-reported use six months post implant was 89% of the nights with an average of 5.8 h every night which is compared very favorably with PAP device use.

Kezirian et al. [19] had very similar results with the use of Apnex Medical Inc. Thirty-one patients with moderate–severe OSA had the device implanted. Follow up polysomnography was conducted at six- and twelve-months post implantation. The AHI was significantly reduced from 45 events per hour at baseline to 21 events per hour at six months. Similarly, other PSG parameters improved at six months (oxygen desaturation index and arousal index). In general, these parameters did not change significantly between six and twelve months. As noticed from other trials, patients with BMI ≥ 35 kg/m^2^ did not respond favorably to HGNS therapy.

Mwenge et al. [47] also had similar results using a different HGNS device (the aura6000 system, ImThera Medical Inc., San Diego, CA, USA). Targeted hypoglossal neurostimulation (THN) uses cyclical neurostimulation to ensure no single nerve fiber is stimulated continuously to avoid muscle fatigue. Unilateral THN was implanted successfully in 13 out of 14 participants. All patients had moderate and severe OSA. Follow up polysomnography was conducted at three and twelve months. There was a significant reduction in AHI from 45.2 events per hour at baseline to 21.7 events per hour at three months and 21.0 events per hour at twelve months. Oxygen desaturation index (ODI) also improved from 36.8 events per hour at baseline to 25 events per hour at three and twelve months. Arousal index was reduced from 37 events per hour to 25 events per hour. Clinically, there was significant improvement in excessive daytime sleepiness at three months and some improvement at twelve months. Fatigue also tended to improve at both three and twelve months.

The third HGNS device that was tested was the Inspire Medical Systems (Maple Grove, MN, USA). It is the only HGNS device so far that is approved by the FDA. Van de Heyning et al. [48] conducted a two-part phase II clinical trial. The first part enrolled 22 patients based on broad selection criteria and all of them underwent Inspire implantation. Overall, the AHI did not change six months post implant compared to baseline (56.1 events per hour vs. 51.1 events per hour, respectively). However, from this phase of the study, the authors determined the factors that were associated with successful response to therapy. The three major factors were baseline AHI, BMI, and the pattern of oropharyngeal collapse. Patients with AHI ≥ 50 events per hour, BMI ≥ 32 kg/m^2^, and complete concentric collapse at the soft palate on DISE exam predicted a poor response to therapy. Accordingly, and after six months, they included nine patients (part 2 of the study) who met all the criteria that predict successful response to therapy and underwent Inspire implantation. After six months, AHI was significantly reduced from baseline at 38.9 events per hour to 10 events per hour after six months. Clinical symptoms (ESS and FOSQ scores) were similar in parts 1 and 2. However, it should be noted that in part 1, the Inspire was implanted to activate the whole hypoglossal nerve trunk, compared to part 2, where it activated the medial branch of the hypoglossal nerve to guarantee stimulation of the protrudor muscles only. No arousals were noted with stimulation. The Stimulation Therapy for Apnea Reduction (STAR) trial is the landmark study of HGNS and it is discussed thoroughly in the next section.

## 7. The Stimulation Therapy for Apnea Reduction (STAR) Trial

A phase III trial for the stimulation therapy for apnea reduction (STAR) was published by Strollo et al. [49] in 2014 and was the first trial that tested the long-term efficacy (12 months) of UAS (Inspire) in a single group of participants and led to FDA approval of the Inspire device. One hundred twenty-six participants were enrolled and had the Inspire implanted. All participants had an AHI between 15 and 65 events per hour and they could not tolerate PAP therapy. Follow up visits were scheduled at baseline (before implantation), and two, six, and twelve months post implant. Polysomnography was conducted at each follow up visit. One hundred twenty-four participants completed the 12 months follow up. The primary outcome of the study was a change in the severity of OSA (reflected by an AHI < 20 events per hour and reduction in the AHI by at least 50% from baseline). The secondary outcome included changes in disease-specific quality of life measures (Epworth Sleepiness Scale (ESS) and the Functional Outcomes of Sleep Questionnaire (FOSQ)). The Inspire settings adjusted during the 12-month period included voltage, rate, and pulse width. The unique feature about the study design is the lack of control group. The participants served as their own controls. After 12 months of enrollment, the first 46 consecutive participants who responded to therapy were randomized into a therapy withdrawal group (where the Inspire was turned off for a week) and a maintenance group (where Inspire treatment was continued) to ensure that the changes over 12 months were related to UAS.

The primary outcome after 12 months showed reduction in the AHI and oxygen desaturation index (ODI) compared to baseline (the median AHI decreased from 29.3 events per hour to 9.0 events per hour (68% reduction, *p* value < 0.001) and the median ODI decreased from 25.4 events per hour to 7.4 events per hour (70% reduction, *p* value < 0.001). The secondary outcome showed an increase in the FOSQ score by more than two points, indicating significant quality of life improvement, and the ESS was <10, indicating no self-reported excessive daytime sleepiness.

Among the 46 responders to therapy, the 23 participants who underwent therapy withdrawal (Inspire was turned off for seven days) had an AHI elevated by an average of 18.2 events per hour (from 7.6 events per hour to 25.8 events per hour, *p* value < 0.001). On the other hand, the maintenance group showed an average increase in AHI by 1.7 events per hour, *p* value < 0.001.

## 8. The Long-Term Outcome of HGNS (The STAR Trial 18, 36, and 60 Months Follow Up)

Participants in the first STAR trial were included in follow up trials to evaluate the long-term effectiveness of the Inspire device. The 18-month post implant trial included 123 participants. The primary and secondary outcomes continue to show stabilization compared to 12 months. The median AHI was 9.7 events per hour (67% reduction compared to baseline), and the median ODI was 8.6 events per hour (67% reduction compared to baseline). The ESS and FOSQ improved significantly compared to baseline. It is also interesting to notice that the stimulation threshold (i.e., the amount of energy in volts needed to activate the tongue movement) remained stable at 18 months compared to one-month post implant. Adherence to device use was similar at 18 months (84%) compared to 12 months (86%) [50].

Stabilization in primary and secondary outcomes was maintained at three years post Inspire implant. A total of 116 STAR trial participants were included in the three-year outcomes trial and 94 of them volunteered to do PSG. Mean AHI was 11.5 events per hour and mean ODI—9.1 events per hour. The group that volunteered in this study did not have significant changes in baseline compared to those who volunteered in the 12-month study. However, the percentage of “non-responders” was lower compared to the 12-month follow up study. One pivotal point that needs to be raised is that even in the same group of responders in this study, the AHI varied, and we still do not understand exactly the importance of this residual AHI. This emphasizes the fact that the AHI is only one metric and it should always be combined with clinical symptoms. ESS and FOSQ scores continued to be stable at 36 months compared to 12 months [51].

After five years of using the Inspire device, there was still good evidence that its effect is sustainable. The five-year outcomes study [18] which enrolled 97 participants of the STAR trial (71 of them volunteered to do PSG) showed that 75% of participants met the surgical success definition (reduction in AHI > 50% of baseline and AHI < 20 events per hour). The response rate to treatment was 63% at 5 years. ESS and FOSQ scores continue to show stabilization compared to 12 and 36 months. However, 67% reported normal FOSQ at 5 years compared to 15% at baseline and 78% reported normal ESS compared to 33%. Self-reported and bedpartner-reported snoring was also stable at five years compared to 12 months. Furthermore, the self-reported use of the device continued to be very good at 80% compared to 86% and 81% at one and three years, respectively. 

We can conclude that PSG parameters and clinical outcomes following the use of Inspire were sustainable long term (up to five years post implant).

Steffen et al. [52] conducted a follow up trial that took place in three ENT centers in Germany using the Inspire system. Sixty patients with moderate–severe OSA who failed PAP therapy were included; 33% of participants had failed either mandibular advancement devices (15 participants) or surgery (14 participants). Follow up was completed at twelve months in 56 out of 60 participants. Participants were screened for enrollment using a home sleep apnea test (HSAT) and DISE exam. The methods and study design were similar to the STAR trial. Baseline AHI was 15–65 events per hour and BMI cut off was 35 kg/m^2^ (compared to 20–50 events per hour and 32 kg/m^2^ in the STAR trial). Since this study used HSAT which tends to underestimate the AHI and since the baseline AHI and BMI in both responders and non-responders were similar, AHI extension to 65 events per hour and BMI cut off to 35 kg/m^2^ was reasonable. Seventy-three percent of participants responded to therapy compared to 66% in the STAR trial and there was significant improvement in quality of life reflected by ESS and FOSQ scores. Even those who did not respond to therapy reported improvement in these quality of life measures. The average adherence at twelve months was 5.6 h per night.

Long-term effectiveness using the ImThera aura 6000 system was conducted by Freidman et al. [20]. They examined 46 patients who had the device implanted and followed up at six months (43/46). In the study, patients were classified into responders and non-responders (responders to HGNS therapy had the following characteristics: BMI < 35 kg/m^2^, baseline AHI < 65, baseline AI ≤ 30 events per hour, and ≤15 events per hour of decreases in SPO2 > 10%). Responders met the primary endpoints which include a statistically significant reduction in AHI and ODI (from 35.7 events per hour at baseline to 8.5 events per hour at six month and from 32.6 events per hour at baseline to 7.9 events per hour at six months, respectively). Responders also had a significant reduction in arousal index from 43.9 events per hour at baseline to 20.4 events per hour at six months, and reduction in ESS from 13 to 8.4.

In summary, all trials demonstrated that HGNS (all devices) is an effective tool in treating moderate to severe OSA in patients who failed PAP therapy. There was a more than 50% reduction in AHI and this response to therapy remained stable after six months. Adherence to therapy was more favorable compared to PAP therapy even after one year. There was also significant clinical response reflected by improvement in excessive daytime sleepiness and fatigue. These studies also demonstrated the essential features to have a good response to HGNS therapy and include a BMI < 35 kg/m^2^, baseline AHI < 65 events per hour, and the absence of complete concentric collapse on DISE exam. HGNS therapy was well tolerated in most patients with minimal adverse events (adverse events will be discussed in detail later in this review). Table 1 summarizes the studies that tested hypoglossal nerve stimulation in adults.

## 9. The Impact of HGNS on OSA-Induced Cardiovascular Morbidities

The literature data evaluating the cardiovascular outcomes following HGNS use are scarce [53]. Since HGNS approval by the FDA in 2014, we do not have sufficient data to evaluate pure cardiovascular end points. Woodson et al. [54] evaluated changes in systolic and diastolic blood pressure readings at 12 and 18 months post implant. Although all participants in this study did not have hypertension, some of them had pre-hypertension with systolic blood pressure (SBP) > 120 and <140 mmHg, and diastolic blood pressure (DBP) > 80 and <90 mmHg. SBP and DBP readings were significantly lower at 12 and 18 months compared to baseline in the therapy-ON group while the therapy OFF group did not have changes in SBP and DBP. Dedhia et al. [53] evaluated heart rate variability in responders in Inspire HGNS from the STAR trial. The analysis included standard deviation of the R–R interval, low-frequency power of the R–R interval, and high-frequency power of the R–R interval. They concluded that responders to Inspire therapy at 12 months had improved heart rate variability. This was the first trial that evaluated a primary cardiovascular outcome. The Cardiovascular Endpoints for Obstructive Sleep Apnea with Twelve Cranial Nerve Stimulation (CARDIOSA-12) [55] is an ongoing trial evaluating several cardiovascular endpoints in patients with OSA who undergoesHGNS implantation. This study is a double-blinded, sham-controlled, randomized, crossover trial. Participants will be randomly assigned to receive active or sham HGNS therapy and then switched into the other intervention group. The primary endpoint is 24-h ambulatory systolic blood pressure (mmHg). Secondary endpoints include 24-h ambulatory diastolic blood pressure (mmHg), nocturnal ambulatory systolic blood pressure (mmHg), nocturnal ambulatory diastolic blood pressure (mmHg), muscle sympathetic nerve activity (MSNA in bursts/min), pre-ejection period (PEP in ms), flow-mediated dilatation (% of baseline), and pulse wave velocity (units/min).

## 10. The Safety of HGNS (Adverse Events and Complications)

Serious adverse events were reported after HGNS implantation. These adverse events can be divided into procedure-related and device-related. In the STAR trial, procedure-related adverse events included postoperative discomfort related to the incision (33 cases, 26%), postoperative discomfort not related to the incision (31 cases, 25%), temporary tongue weakness (23 cases, 18%), and sore throat from intubation (15 cases, 12%), and headache (8 cases, 6%). Device-related adverse events included discomfort secondary to electrical stimulation (50 cases, 40%), tongue abrasion (26 cases, 21%), mouth dryness (13 cases, 10%), functionality issues with the implanted device (13%), and mechanical pain associated with the device (8 cases, 6%). Almost 75% of these procedure-related events were resolved by 18 months post implant. Four patients had serious adverse events. Two were device-related (repositioning of the generator) and two patients died (one secondary to a cardiac event that is thought not to be related to the device and the second was secondary to homicide) [49].

Three to five years post implant, device-related adverse events continued to decrease. The most commonly reported adverse events were tongue discomfort secondary to repetitive stimulation. In most cases, this resolved by adjusting the programming parameters and sometimes by dental adjustment. The frequency of this adverse event decreased from 81 times after one year to 5 times after five years. Tongue abrasion reporting was decreased from 28 times after one year to 2 times after five times. Only one participant (after five years) underwent repositioning of the stimulation leads to alleviate tongue discomfort. As mentioned earlier, the stimulation energy needed to activate the tongue was not increased after five years, proving that there was no fibrosis between the stimulating lead cuff and the nerve fibers [18,51].

Bestourous et al. [56] reviewed the medical device-related adverse events reported in the manufacturer and user facility device experience (MAUDE) database. These events are reported to the FDA and they are mandatory for the manufacturer (Inspire) and optional for the patients and health care providers. One hundred eighty reports, containing a total of 196 adverse events, were reviewed between July 2014 and April 2020. These adverse events were classified into (a) patient-related, (b) device-related, and (c) iatrogenic. The most common patient-related adverse events were infection (50 cases, 34%), neuropraxia (22 cases, 15%) and seroma/hematoma (17 cases, 12%). Other adverse events included pain (13 cases, 9%), device expulsion through skin (9 cases, 6%), device migration (12 cases, 8%), tongue swelling (1 case, 0.7%), neck swelling (3 cases, 2%), sialorrhea (1 case, 0.7%), and muscle tethering or traction (6 cases, 4%). The most common device-related adverse events were related to faulty placement (10 cases, 35%) and sensing leads malfunction (10 cases, 35%). Others include stimulation lead malfunction (2 cases, 7%), generator malfunction (2 cases, 7%), and remote-control malfunction (3 cases, 10%). The most common iatrogenic events were pleural injury with or without pneumothorax (13 cases, 62%). Other iatrogenic adverse events included vascular injury (6 cases, 29%), musculoskeletal injury (1 case, 5%), and mucosal injury (1 case, 5%). A meta-analysis published by Certal et al. showed an infection rate of 0.5% [57].

Rare cases of internal or external device malfunction include device migration, as well as salivary flow changes, lip weakness, and paresthesia [58]. Allergic reaction, hypertonicity, wound dehiscence, device expulsion through the skin, and muscle or lead tethering were also reported [56]. In a follow up study comparing adults over age 65 who generally have more comorbidities to a matched control group under age 65, surgical times were similar. Remarkably, no patient had perioperative respiratory, cardiac, or neurologic adverse events. Neither group required care in the ICU or repeated intubation after surgery [25]. No mortalities directly related to HGNS were reported so far; although one patient experienced cardiac arrest in the operating room after device placement, he was resuscitated and recovered in the ICU [56].

## 11. Hypoglossal Nerve Stimulation in Children

All published data to date on the use of HGNS for OSA have focused on children with Down syndrome (DS). Children with Down syndrome are known to be at very high risk for obstructive sleep apnea. Prior research has shown that the prevalence of OSA in infants with DS is 31% [59], and by mid-childhood, this increases to 66% [60]. OSA has been associated with neurocognitive impairment and impaired health-related quality of life in children with DS [61,62,63]. Current treatments for OSA in children with DS include adenotonsillectomy and positive airway pressure (PAP) therapy. Unfortunately, adenotonsillectomy may be ineffective in up to 73% of children with DS [64,65]. Similarly, PAP therapy, while efficacious, is limited by poor adherence, as only 46% of children with DS prescribed PAP therapy for OSA are adherent to therapy [66]. Given the lack of a current effective treatment combined with the high prevalence of OSA in children with DS, there is a large unmet need for an effective therapy for OSA in this population.

Hypotonia is a cardinal feature of DS, including airway hypotonia during sleep. This airway hypotonia during sleep has been identified as a cause of OSA in children with DS, as well as a cause of persistent OSA after adenotonsillectomy [67]. Given this hypotonia, OSA treatment aimed at improving airway tone is particularly well-suited to children with DS. As HGNS directly targets improving airway muscle tone, children with DS appear to be an ideal population for this modality of treatment.

All six published manuscripts to date on the use of HGNS in children have been based off a single ongoing clinical trial (NCT02344108, “A Pilot Study to Evaluate the Hypoglossal Nerve Stimulator in Adolescents with Down Syndrome and Obstructive Sleep Apnea”). Of these six manuscripts, three have focused on hypoglossal nerve stimulator efficacy results [68,69,70], while two discussed the surgical approach to hypoglossal nerve stimulator implantation [71,72] and one reported the anesthesia approach to hypoglossal nerve stimulator implantation [73].

### 11.1. Efficacy of Hypoglossal Nerve Stimulation in Children with Down Syndrome

The manuscripts published to date on the efficacy of hypoglossal nerve stimulation in children with DS have shown promising results. The first published result was a case study in a 14-year-old boy with severe OSA and CPAP intolerance resulting in need for a tracheostomy [68]. The patient’s baseline AHI with capped tracheostomy tube showed an AHI of 48.5 events/hour. After device implantation and titration, his AHI was reduced to 3.4 events/hour (93% improvement in AHI). Given this, his tracheostomy was removed, and he was able to use HGNS alone for his OSA.

A subsequent case series was then published in 2018 [69]. This series reported the first six adolescents (age range 12–18 years old) with DS to receive the hypoglossal nerve stimulator. In this group, the average baseline AHI was 26.4 events per hour, which improved to a residual AHI of 5.0 events per hour in follow up 6–12 months post implant. There was also improvement in OSA-18 scores, a pediatric-specific OSA health-related quality of life metric. Device usage per night was an average of 8.8 h, with a minimum of 5.6 h per night. It was noted that two participants required hospital readmission after surgery, one for a possible chest incision site cellulitis as well as one admitted due to uncontrolled pain as an outpatient. Both participants recovered and were able to remain in the study without further complications reported.

The most recent report presented preliminary results for the first twenty participants in the clinical trial [70]. The results presented in this study included the participants from the prior case report as well as case series. Median baseline AHI was 24.2 events per hour, which improved to 3 events per hour at two months after device titration. This 87.6% reduction in median AHI is greater than the 68% reduction in median AHI reported in the STAR trial [18], consistent with the idea that targeting airway hypotonia in children with DS may be particularly effective. In addition to the improvement in AHI, quality of life as measured by the OSA-18 also improved with a moderate effect size. Participants were noted to have excellent adherence (median use of 9.2 h per night).

### 11.2. Pediatric Modifications to Hypoglossal Nerve Stimulator Implantation

In general, the pediatric inclusion criteria used in the current clinical trial closely reflect the criteria originally used in the STAR trial with a few modifications. Similar to the STAR trial, circumferential collapse of the velopharynx and obesity were among the exclusion criteria. Due to age-based norms for obesity in children, obesity was defined as a BMI >95th percentile for age and gender rather than an absolute BMI cutoff. Due to the inclusion of children with Down syndrome and associated cognitive impairment, parents were required to attest that their child could communicate discomfort as well as cooperate with study exams.

Device implantation is similar to that for adults, with slight modifications. In children, only a single incision is used, and a tunneled approach is used to place the implantable pace generator and respiratory sensing lead [71,72]. Additionally, due to smaller patient size and associated decreased amount of space available in the chest, there is a concern about possible cardiac interference with the sensing lead. To prevent this, the sensing lead is placed laterally from the incision rather than medially as is done in adults to increase the distance from the heart.

A single case series has reported the anesthesia experience with HGNS in children with DS [73]. The authors reported using inhalational induction for anesthesia as well as extubation from deep anesthesia with close monitoring and available non-invasive ventilation. They reported the procedure was well-tolerated in their small sample with no postoperative complications.

Overall, the existing data in children with Down syndrome suggest that hypoglossal nerve stimulation is well-tolerated and effective for the treatment of OSA in children with Down syndrome. Future studies are needed to extend these results to children without Down syndrome as well. Additionally, results to date are generally from adolescents, with a minimum age of 10 years for study participation. Future studies will be needed to extend the use of hypoglossal nerve stimulation for younger children. Given the large amount of growth from infancy to early childhood, there may be a minimum age or size requirement for safe implantation of a hypoglossal nerve stimulator. It is possible that in addition to the need to replace the device battery approximately every ten years, there may be a need for interval procedures to adjust leads due to growth in younger children.

## 12. The Financial Cost of Hypoglossal Nerve Stimulation

From a financial point of view, HGNS is an expensive procedure. In one report, for patients who meet the STAR trial inclusion criteria, the estimated lifetime incremental cost effectiveness ratio (ICER) was USD39,471 per quality-adjusted life year (QALY). Although this is still below the currently accepted ICER in the US of USD40–50K/QALY, it is much more costly than PAP therapy (ICER of USD15,915/QALY). With more insurance plans covering the procedure, still, the out-of-pocket and co-pays for the procedure and even for diagnostic workup before the procedure (such as DISE exam) might be financially overwhelming for many patients.

## 13. The Future of Hypoglossal Nerve Stimulation (What Is New?)

Since the early 1970s when HGNS was used in animal trials, the system of hypoglossal nerve stimulation has developed significantly over the last fifty years. With each new system, the goal was to ensure maximum patient safety, avoid serious adverse events, and minimize local side effects to guarantee the best adherence and tolerance. In September 2019, Eastwood et al. [74] introduced a new device for HGNS called the Genio system (Nyxoah SA, Mont-Saint-Guibert, Belgium) (Figure 11). It has the same principle of the previous HGNS devices with some modifications. (A) The Genio system is based on bilateral hypoglossal nerve stimulation instead of unilateral stimulation [20]. The benefit from this is to ensure more forward tongue protrusion compared to possible right or left deviation with unilateral stimulation [75]. (B) The Genio system uses paddle electrodes that are inserted around the most distal hypoglossal nerve branches close to its insertion in the genioglossus muscle [76]. This guarantees pure stimulation of the genioglossus muscle compared to previous HGNS devices that used cuff electrodes wrapped around the more proximal hypoglossal nerve branches, causing possible stimulation of other tongue muscles [77,78]. (C) The Genio system has a small stimulation unit that does not contain batteries. This potentially results in a smaller incision, shorter surgery time, and faster healing. The implanted stimulation unit receives energy pulses from an external activation unit which is placed under the chin and attached to an adhesive disposable patch. This eliminates the need for a stimulation lead, cuff electrodes, and the generator. (D) The Genio system provides cyclical pauses which allow rest periods in between breathing cycles. Stimulation is not synchronized to inspiration, as is the case in the Inspire system, which eliminated the need for a sensing lead [46,49].

There were no device-related serious adverse events (SAE) six months post implant. The most common device-related non-SAE was local skin irritation due to the adhesive patch which resolved in all participants except one, tongue abrasion (11%), and fasciculation (11%). Four procedure-related SAE were reported: three of them were related to local infection requiring explantation and the fourth one was impaired swallowing that resolved without any intervention. Procedure-related non-SAE included painful swallowing (30%), dysarthria (26%), hematoma (19%), and bruising around the incision site (19%).

Mean AHI was decreased from 24 events per hour at baseline to 13 events per hour at six-months. Mean ODI was decreased from 19 events per hour to 10 events per hour. Clinically, the outcome was favorable where ESS dropped from 11 to 8, sleep efficiency improved, and bedpartner-reported loud snoring was decreased from 96% to 35%.

## 14. Conclusions

Hypoglossal nerve stimulation seems to be a novel therapy for moderate to severe OSA patients who cannot tolerate PAP therapy. With the role of OSA as a risk factor for many chronic systemic diseases and in the context of extremely variable and challenging adherence to PAP therapy, HGNS emerges as a very promising therapy with very good long-term effectiveness and adherence. HGNS is very promising in children with OSA and Down Syndrome which is another group of patients in whom PAP adherence is extremely challenging.

On the other hand, HGNS is still an invasive procedure with the associated risks involved in an invasive procedure. Although and as mentioned earlier, HGNS systems keep developing to minimize adverse events and enhance safety. Moreover, HGNS is not an optimal alternative option for all patients with moderate–severe OSA (especially patients with BMI ≥ 32 kg/m^2^). 

## Figures and Tables

**Figure 1 ijerph-18-01642-f001:**
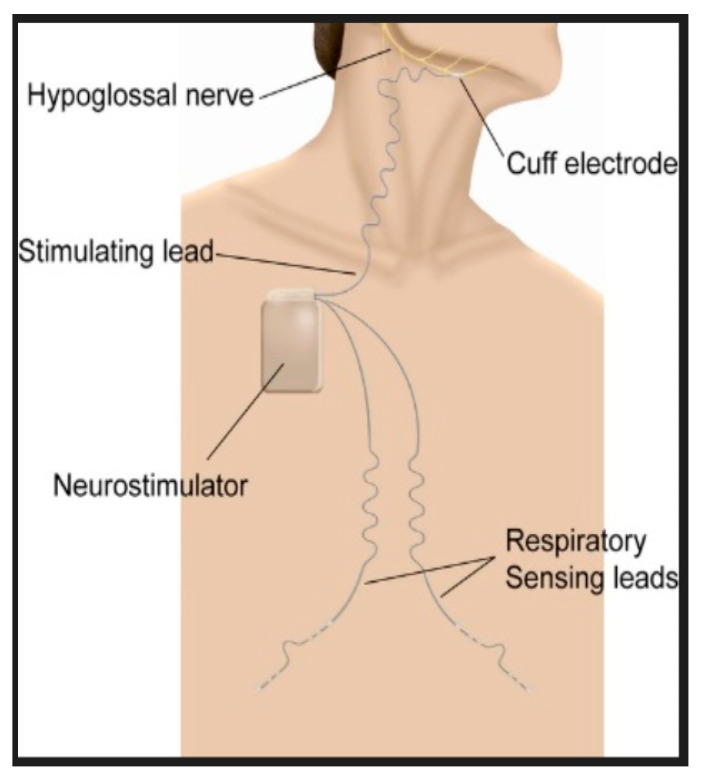
Apnex Medical Inc. (St. Paul, MN, USA).

**Figure 2 ijerph-18-01642-f002:**
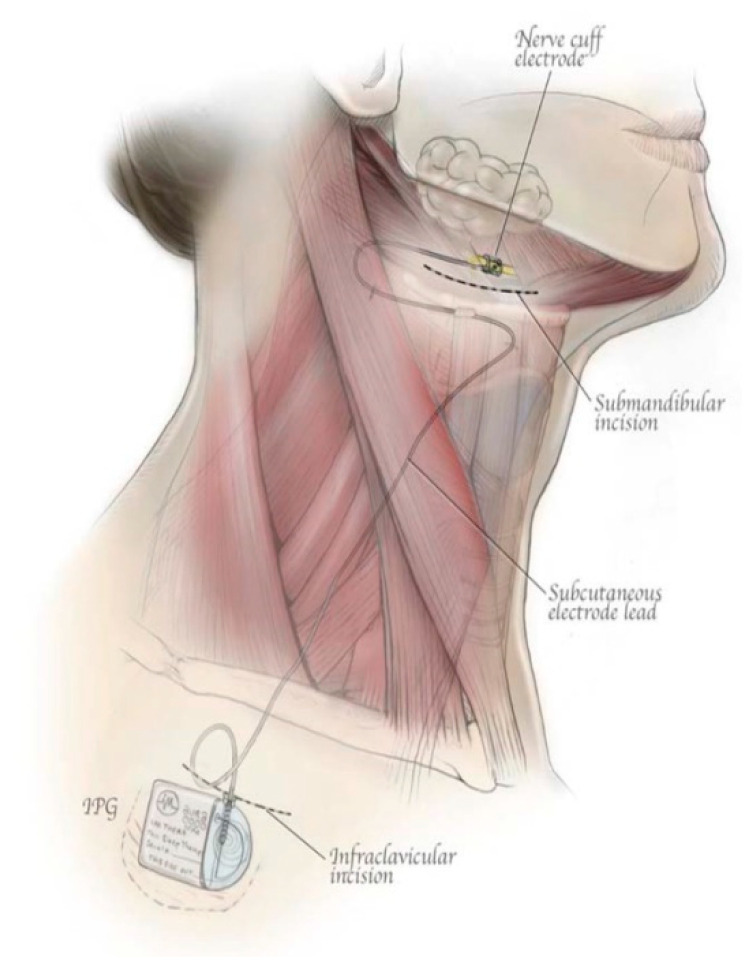
ImThera aura6000 (San Diego, CA, USA).

**Figure 3 ijerph-18-01642-f003:**
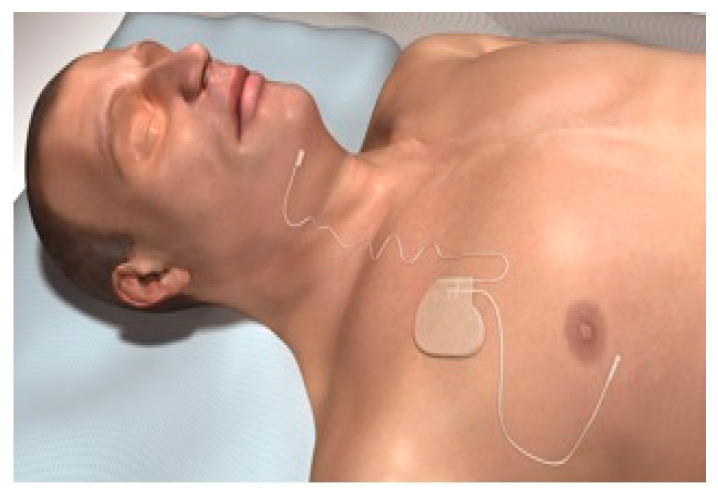
The Inspire system (Maple Grove, MN, USA).

**Figure 4 ijerph-18-01642-f004:**
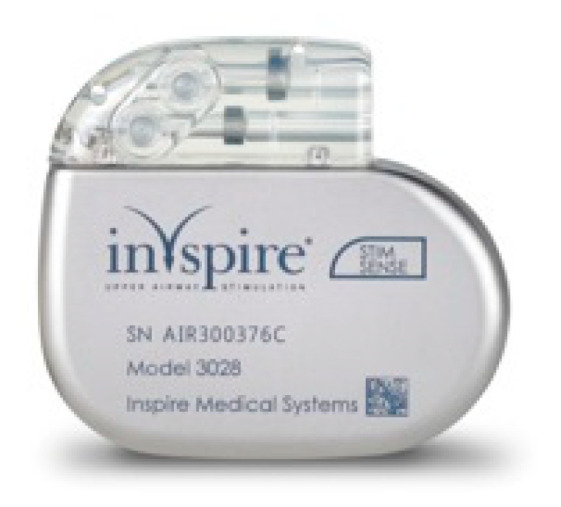
The Inspire generator.

**Figure 5 ijerph-18-01642-f005:**
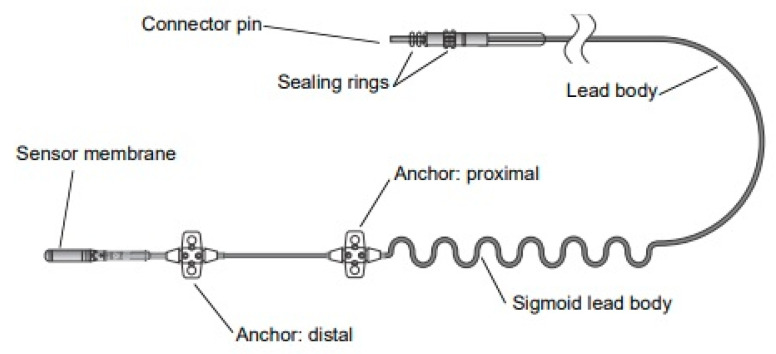
The Inspire respiratory sensing lead.

**Figure 6 ijerph-18-01642-f006:**
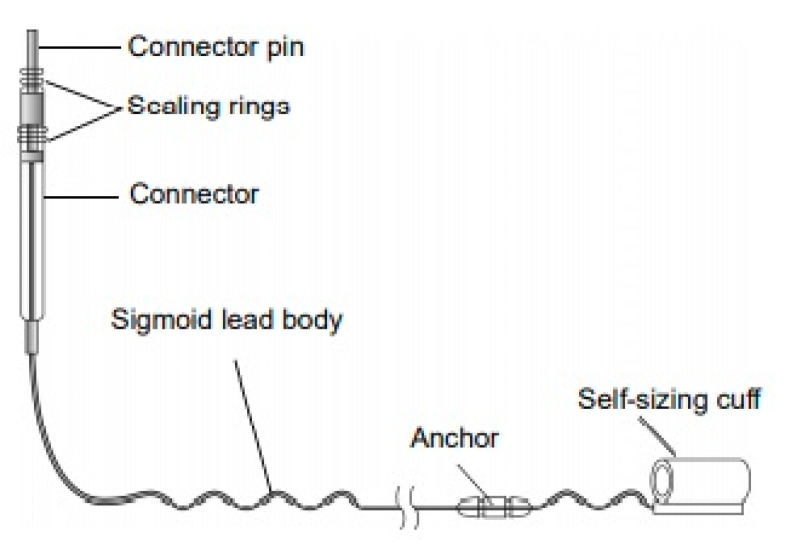
The Inspire stimulation lead.

**Figure 7 ijerph-18-01642-f007:**
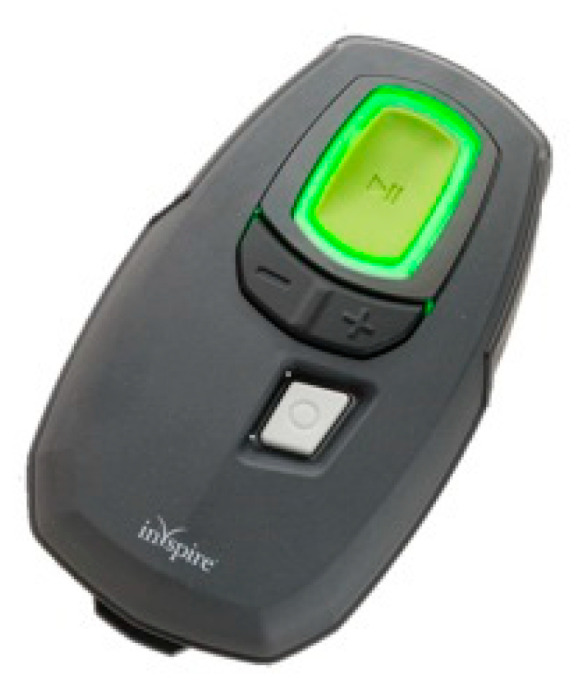
The Inspire remote.

**Figure 8 ijerph-18-01642-f008:**
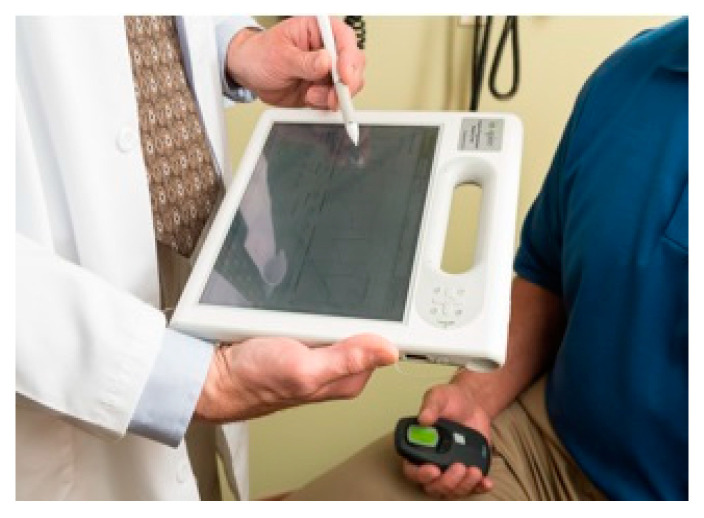
The Inspire programmer.

**Figure 9 ijerph-18-01642-f009:**
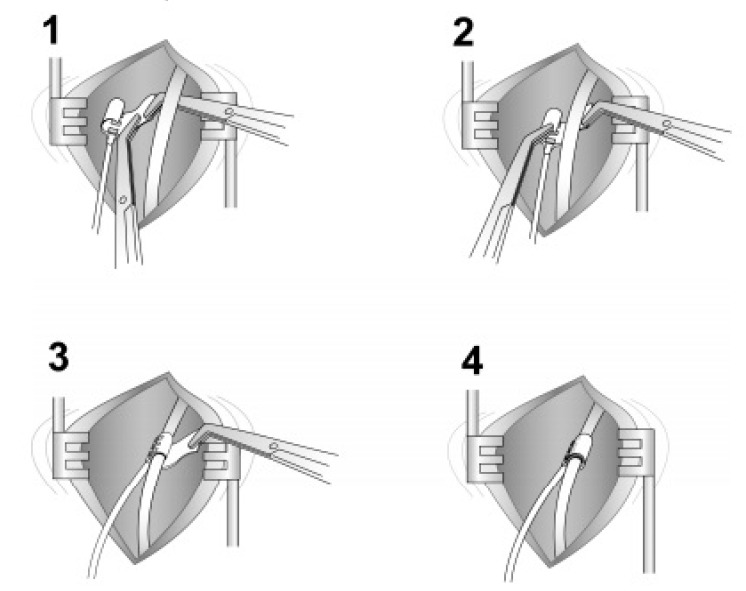
The neck incision (placing the cuff around the hypoglossal nerve). The steps of placing the cuff around the hypoglossal nerve 1–4.

**Figure 10 ijerph-18-01642-f010:**
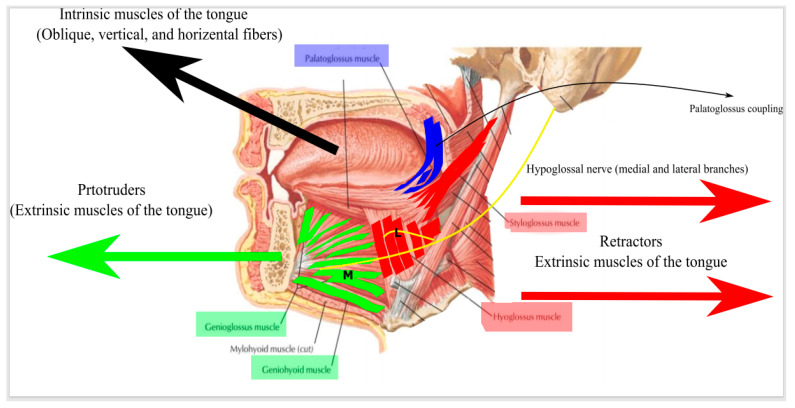
A sagittal section in the head showing the intrinsic and extrinsic muscles of the tongue and their role in the protrusion and retraction of the tongue in addition to the opening of the retropalatal and hypopharyngeal space via palatoglossus coupling. (L) and (M) are the lateral and medial branches of the hypoglossal nerve, respectively.

**Figure 11 ijerph-18-01642-f011:**
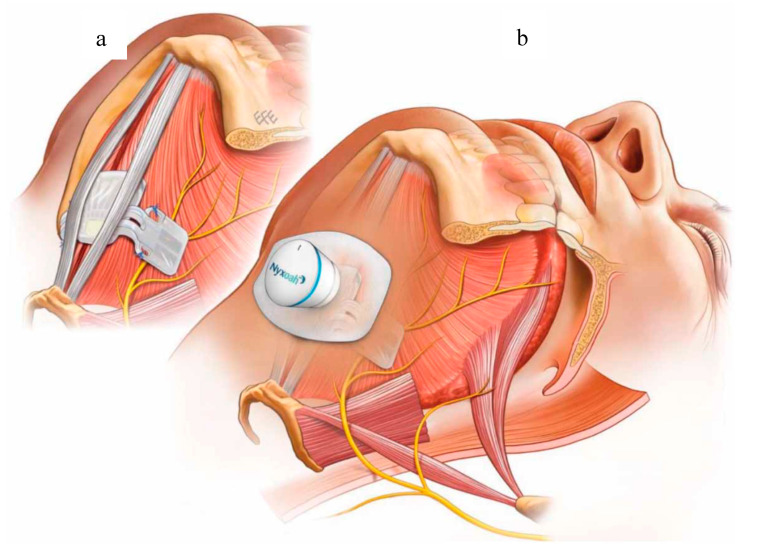
The Genio system. (**a**) The implanted stimulator straddling the genioglossus muscles and hypoglossal nerve branches bilaterally and (**b**) the disposable patch and activation unit.

**Table 1 ijerph-18-01642-t001:** Clinical trials conducted using hypoglossal nerve stimulation in adults.

Author, Year	(UAS) Device	Unilateral/Bilateral HGNS	Sample	Follow Up (m)	mAHI (before)	mAHI (after)	Other PSG Parameters Changes	Clinical & Functional Changes
Schwartz et al., 1996	- Direct intramuscular stimulation of lingual musculature	N/A	9	N/A	65	9	N/A	N/A
Oliven et al., 2007	- Surface GG muscle stimulation. - Surface GG + HG coactivation	N/A	7	N/A	40	- No ∆ Pn/F, No ∆ OP CSA - ↓ Pn/F, ↑ OP CSA	N/A	N/A
	- Intramuscular GG stimulation - Intramuscular GG+HG stimulation		7		38	- ↓ Pn/F, ↑ OP CSA - No additional dilation		
Eastwood et al., 2011	Apnex Medical Inc.	Uni	21	6	43	20	- ↓ N1 - ↑ REM - ↑ SPO2 - ↓ arousals	- Improvement in quality of life questionnaires. - ↓ ESS
Kezirian et al., 2014	Apnex Medical Inc.	Uni	31	6 & 12	45	21 (6 m)	- ↓ N1 - ↑ REM - ↑ SPO2 - ↓ arousals - ↑ SE	- Improvement in quality of life questionnaires. - ↓ ESS
Mwenge et al., 2013	ImThera Medical Inc.	Uni	13	3 & 12	45	22	- ↓ ODI - ↓ arousals	- Improvement in quality of life questionnaires. - ↓ ESS
Van de Heyning et al., 2012	Inspire (phase II clinical trial) *Part II of the trial (after determining the responders to Rx)	Uni	9	6	39	10	- N/A	- No ∆
Strollo et al., 2014	Inspire (STAR trial)	Uni	126	12	29	9	- ↓ ODI	- Improvement in quality of life questionnaires. - ↓ ESS
Strollo et al., 2014	Inspire (STAR trial 18 m follow up)	Uni	123	18	29	10	- ↓ ODI	- Improvement in quality of life questionnaires. - ↓ ESS
Woodson et al., 2016	Inspire (STAR trial 3 y follow up)	Uni	116	36	29	12		- Improvement in quality of life questionnaires. - ↓ ESS
Woodson et al., 2018	Inspire (STAR trial 5 y follow up)	Uni	97	60	29	12	- ↓ ODI	- Improvement in quality of life questionnaires. - ↓ ESS
Steffen et al., 2018	Inspire	Uni	60	6 & 12	31	14 (1 year)	- ↓ ODI	- Improvement in quality of life questionnaires. - ↓ ESS
Freidman et al., 2016	ImThera Medical Inc.	Uni	46	6	36	9	- ↓ ODI - ↓ arousals	- ↓ ESS
Eastwood et al., 2019	Genio system	Bilateral	22	6	24	13	- ↓ ODI	- ↓ ESS - ↓ loud snoring

UAS = upper airway stimulation; HGNS = hypoglossal nerve stimulation; mAHI = mean apnea-hypopnea index; PSG = polysomnography; Pn/F = nasal pressure/flow ratio; OP = oropharyngeal; CSA = cross-sectional area; SPO2 = oxygen saturation; ESS = epworth sleepiness scale; GG = genioglossus; HG = hyoglossus; SE = sleep efficiency; ODI = oxygen desaturation index; STAR = The Stimulation Therapy for Apnea Reduction.

## Data Availability

Not applicable.

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
