# Peer review of "The Hypoglossal Nerve Stimulation as a Novel Therapy for Treating Obstructive Sleep Apnea—A Literature Review"

_ijerph, 2021, doi:10.3390/ijerph18041642_

Round 1

Reviewer 1 Report

Thank you for the opportunity to review this paper. Overall, I find it to be a very well-written and thorough review of an important topic, and I believe it will become a valuable source of information for clinicians wanting to know more about hypoglossal nerve stimulation therapy.

Major comments:

  • consider a summary table of the 31 articles included in the review (perhaps as supplementary information). This could summarise: year of publication, comparison/research question, method, main outcome and an assessment of study quality 
  • Section 3 of the review is the one I have the most problem with. This reads to me like a user manual for the Inspire device which i think is outside the scope of this literature review. Please consider simplifying this section. There are too many figures. I would remove Figures 8 and 8a-8e and avoid providing instructions on how to use the device. This is not a summary of the literature and readers can find this information elsewhere.
  • More critique of the literature is required. Your summary of the main studies (section 6,7,8) is excellent but there is little mention of any limitations nor recommendations for future research. 
  • Section 10 - adverse events are "downplayed". You report two serious events in the STAR trial (out of ??124). This is NOT rare. According the to WHO it is considered "common" (1/10 to 1/100). Furthermore, the sample size is very small and so the confidence intervals around these rates would be large. What were the two serious events - death?, permanent impairment? etc) Is there any more information about them?
  • Later in section 10 - do you know what the denominator is for the 196 adverse events reported?(ie how many clinical device implants were there at that time?). Can you please provide the whole number with the % of different event types? 
  • Section 13 - Conclusion. There are two new concepts introduced in the conclusion - type 2 diabetes and OSA, and the costs of HGNS. They need to be introduced and discussed earlier in the review to warrant a mention in the conclusion. Please consider removing the reference to type 2 diabetes in the conclusion, and adding another section to the review on the costs of the implant.

Minor comments:

  • Line 31 - this is a very old reference. Prevalence rates are estimated to be much higher now. There have been several more large prevalence studies and a systematic review of prevalence published in 2017 estimating rates of up to 38%.
  • Section 4 - you discuss pre-op procedures here as well so please consider changing the title to "implementation process and pre-operative and post-operative procedures". 
  • Line 327 - this is the first reference to the STAR trial. Please signpost readers to the following section (ie"...the landmark study of HGNS and discussed in the next section...")

Author Response

Dear Reviewer 1

  Thank you very much for the feedback and appreciate the valuable comments. I responded to all comments as below:

Major comments:

  • consider a summary table of the 31 articles included in the review (perhaps as supplementary information). This could summarise: year of publication, comparison/research question, method, main outcome and an assessment of study quality --- A summary table of clinical trial used hypoglossal nerve stimulation devices in adults is added

  • Section 3 of the review is the one I have the most problem with. This reads to me like a user manual for the Inspire device which i think is outside the scope of this literature review. Please consider simplifying this section. There are too many figures. I would remove Figures 8 and 8a-8e and avoid providing instructions on how to use the device. This is not a summary of the literature and readers can find this information elsewhere ----- Edited, removed the figures (8 and 8a-8e) and removed many of the details that can be read in the manual.

  • More critique of the literature is required. Your summary of the main studies (section 6,7,8) is excellent but there is little mention of any limitations nor recommendations for future research--- Added more details regarding adverse events which should add a balance to the review (benefits and adverse events)

  • Section 10 - adverse events are "downplayed". You report two serious events in the STAR trial (out of ??124). This is NOT rare. According the to WHO it is considered "common" (1/10 to 1/100). Furthermore, the sample size is very small and so the confidence intervals around these rates would be large. What were the two serious events - death?, permanent impairment? etc) Is there any more information about them?---Added the detailed adverse events in STAR trial and the serious adverse events.
  • Later in section 10 - do you know what the denominator is for the 196 adverse events reported?(ie how many clinical device implants were there at that time?). Can you please provide the whole number with the % of different event types? --- Unfortunately, I could not find the denominator. They just listed the period of time. Added more details related to all adverse events (number of cases and %)

  • Section 13 - Conclusion. There are two new concepts introduced in the conclusion - type 2 diabetes and OSA, and the costs of HGNS. They need to be introduced and discussed earlier in the review to warrant a mention in the conclusion. Please consider removing the reference to type 2 diabetes in the conclusion, and adding another section to the review on the costs of the implant---- Removed the financial cost to a separate paragraph and removed REM-OSA and Type-II DM.

Minor comments:

  • Line 31 - this is a very old reference. Prevalence rates are estimated to be much higher now. There have been several more large prevalence studies and a systematic review of prevalence published in 2017 estimating rates of up to 38% -- Added a new reference

  • Section 4 - you discuss pre-op procedures here as well so please consider changing the title to "implementation process and pre-operative and post-operative procedures"---Edited

  • Line 327 - this is the first reference to the STAR trial. Please signpost readers to the following section (ie"...the landmark study of HGNS and discussed in the next section...")---Edited

Reviewer 2 Report

In "The Hypoglossal Nerve Stimulation as a Novel Therapy for Treating
Obstructive Sleep Apnea – A literature Review", the authors have
provided a literature review of hypoglossal nerve stimulation (HGNS) or
upper airway stimulation (UAS) as alternatives to continuous positive
airway pressure (CPAP). The authors conculude that hypoglossal nerve
stimulation is superior to CPAP for the treatment of obstructive sleep
apnea in patients who have a difficult time tolerating CPAP. They also
find that adherence to HGNS is superior to adherence to CPAP. The
manuscript is well-written and comprehensive in its review of
hypoglossal nerve stimulation as a promising therapy for OSA,
particularly in subjects with moderate-to-severe obstructive
apnea. The authors also include coverage of trials in pediatric
subjects and this is a welcome addition in this kind of review. I have
no major concerns with the manuscript and I appreciate the authors'
comprehensive coverage of this very interesting therapy. I have a few
minor concerns that I have listed below and would appreciate the
authors' addressing these issues.

Minor concerns
1. Throughout. Rather than referring to "the hypoglossal nerve
stimulation" the authors can usually omit the article and just use,
"hypoglossal nerve stimluation."

2. Line 200 (pg. 9). It appears that reference 31 is not formatted
corrrectly here.

3. Line 232. Add "a" to make "...which are a group of horizontal..."

4. Line 255. "result" should be "results".

5. Section on human studies. It would be very helpful if the authors
could summarize these results in a table since it is difficult to keep
the results straight when just reading through the text. The table
could include summary information about how many subjects were tested
and what the main finding was for each stimulation location/method.

6. Line 377. Why not just say, "All participants had an AHI between 15
and 65 events per hour...." It seems much more straight-forward.

7. Line 473. What metric of heart-rate variability did Dedhia et
al. use? I realize we have the original reference cited but, as a
convenience to the reader, it would be nice to have the specific
metric mentioned here.

8. Line 503. Omit "feeling".

9. Line 520 seems to end in an incomplete sentence.

10. Line 645. Omit "a" from "a very good long-term..."

11. Line 653. "On the other hand..." would sound more natural than "On
the other side..." or perhaps, "In contrast..."

Author Response

Dear Reviewer 2

  Thank you very much for the feedback and appreciate the valuable comments. I responded to all comments as below:

Minor concerns

  1. Throughout. Rather than referring to "the hypoglossal nerve
    stimulation" the authors can usually omit the article and just use,
    "hypoglossal nerve stimluation."---Edited, removed “the” from almost all HGNS

  1. Line 200 (pg. 9). It appears that reference 31 is not formatted
    corrrectly here--- Edited and formatted correctly

  1. Line 232. Add "a" to make "...which are a group of horizontal..."---Edited

  1. Line 255. "result" should be "results"---Edited

  1. Section on human studies. It would be very helpful if the authors
    could summarize these results in a table since it is difficult to keep
    the results straight when just reading through the text. The table
    could include summary information about how many subjects were tested
    and what the main finding was for each stimulation location/method --- A table is added

  1. Line 377. Why not just say, "All participants had an AHI between 15
    and 65 events per hour...." It seems much more straight-forward---Edited

  1. Line 473. What metric of heart-rate variability did Dedhia et
    al. use? I realize we have the original reference cited but, as a
    convenience to the reader, it would be nice to have the specific
    metric mentioned here --- The metric is added

  1. Line 503. Omit "feeling"--- Edited

  1. Line 520 seems to end in an incomplete sentence --- Edited

  1. Line 645. Omit "a" from "a very good long-term..."---Edited

  1. Line 653. "On the other hand..." would sound more natural than "On
    the other side..." or perhaps, "In contrast..."---Edited

Round 2

Reviewer 2 Report

I wish to thank the authors for taking my suggestions to heart and revising the manuscript. I think it is a far stronger work now and provides and excellent and comprehensive review of hypoglossal nerve stimulation as a therapy for OSA.